# The Measurement of Elderly Volunteers’ Optic Nerve Sheath Diameters by Ocular Ultrasonography

**DOI:** 10.3390/medicina55080413

**Published:** 2019-07-27

**Authors:** Mustafa Avci, Nalan Kozaci, Erdal Komut, Seval Komut, Gulsum Caliskan, Gul Tulubas

**Affiliations:** 1Department of Emergency Medicine, University of Health Sciences, Antalya Education and Research Hospital, 07100 Antalya, Turkey; 2Department of Radiology, Hitit University Medical School, 19040 Corum, Turkey; 3Department of Emergency Medicine, Hitit University Medical School, 19040 Corum, Turkey

**Keywords:** elderly population, emergency service, hospital, intracranial pressure, ocular ultrasonography, optic nerve sheath diameter, ONSD

## Abstract

Background and objectives: The optic nerve is a component of the central nervous system, and the optic nerve sheath is connected to the subarachnoid space. For this reason, intracranial pressure (ICP) increases are directly transmitted to the optic nerve sheath. Knowing the normal optic nerve sheath diameter (ONSD) range in a healthy population is necessary to interpret this measurement as a sign of intracranial pressure in clinical practice and research. In this study, we aimed to determine the standard ONSD value in healthy adultsaged65 years of age or older who had not previously been diagnosed with a disease that could increase the ICP. Materials and Methods: The right and left ONSD values and ONSD differences were compared, according to the gender of the patients. The patients were divided into 3 groups, according to their age. The age groups were assigned as follows: Group 1: 65–74 years of age; Group 2: 75–84 years of age; and Group 3: 85 years of age or older. The ONSDs and the ONSD difference between the left and right eyes of Group 1, Group 2 and Group 3 were compared. Results: The study included 195 volunteers. The mean ONSD of both eyes was 4.16±0.69 mm, and the difference between the ONSD of the left and right eyes was 0.16±0.18 mm. There was no difference between genders in terms of right ONSD, left ONSD, mean ONSD and ONSD difference between the left and right eyes. There was no correlation between age and ONSD and ONSD difference. When the age groups and ONSD were compared, no difference was found between the groups. Conclusions: In conclusion, the ONSDs of both eyes do not vary with age in healthy adults aged65 years or older. ONSD does not vary between genders. The calculation of ONSD difference can be used to determine ICP increase.

## 1. Introduction

Aging can be defined as gradual changes in the body structure that are not caused by any disease or trauma [1]. Numerous studies have shown that significant age-related changes occur in the brain. It has been shown that, as age increases, the weight or volume of the brain decreases, the amount of cerebrospinal fluid increases, and the volumes of the temporal lobe, cerebellar vermis, cerebellar hemispheres, hippocampus and prefrontal white matter decrease [1,2,3].

Intracranial pressure (ICP) is the pressure caused by the total brain volume, cerebrospinal fluid (CSF) and blood in the skull, and it is stable. ICP increases as a result of any increase in the volume of the intracranial structures due to conditions, such as head trauma, ischemic stroke, hemorrhagic stroke, the presence of mass and infection [4]. The measurement of ICP is useful in the aforementioned clinical conditions, because high intracranial pressure indicates a severe pathology and requires rapid treatment [5]. The most precise method of measuring ICP is a direct invasive measurement of the intraventricular or subdural pressure. This invasive method is not practical in emergency departments. It also carries the risk of intracerebral hemorrhage and infection [5]. ICP can be non-invasively measured by a computed tomography (CT) scan; however, only the secondary characteristics of increased ICP can be visualized. Nowadays, CT scans are readily available in many hospitals, but in many cases, they are still not easily accessible, and the transfer of patients can also pose a problem [6,7]. Therefore, the measurement and monitoring of ICP should be performed using a non-invasive, simple, reproducible and bedside method, especially for emergency department patients [4].

The optic nerve is a component of the central nervous system, and the optic nerve sheath is connected to the subarachnoid space. For this reason, an intracranial pressure increase is directly transmitted to the optic nerve sheath [8].

Since ultrasonography (USG) is easy to learn and is a rapid, bedside, non-invasive, portable and reproducible method, it has become an irreplaceable component of emergency departments [4,5,6]. Recently, the measurement of the optic nerve sheath diameter (ONSD) by ocular USG has been widely used to evaluate ICP increase. The studies have shown that ONSD increases in cases leading to increased intracranial pressure. It has also been shown that ONSD can be used in the early diagnosis of intracranial hypertension in patients with brain death and to estimate the increased risk of intracranial pressure due to traumatic brain injury in a prehospital setting [4,8,9,10,11]. However, it may not always be possible to perform ONSD measurement with full accuracy. In addition, ONSD measurements may vary, depending on the participant performing the measurements. Since ONSD measurements are made at a millimeter scale, very small measurement differences may alter the results [12]. Therefore, in order to make reliable measurements, a standardized measurement method, which is less affected by personal errors but still requires some skill, should be used [9,12,13]. For this purpose, the use of A-scan and B-scan techniques in ONSD measurement was evaluated [13,14].

The studies on ONSD have shown that there are significant differences between individuals [15], but there is no difference between boys and girls [16]. While 5 mm is the most commonly used cut-off value for adults, values of up to 5.9 mm were used in different studies [15]. It has been shown that ONSD increases with age in children, and that the rate of increase in the first years of life is the highest [15,16,17].

Knowing the normal ONSD range in a healthy population is necessary to interpret this measurement as a sign of intracranial pressure in clinical practice and research. The standard ONSD value of the elderly population (65 years and older) may vary due to atrophy, which may develop in the cerebral structures, as a result of the aging process. The early detection of an intracranial pathology that develops in such a case may be difficult. Therefore, there is a need for defining a standard ONSD for the elderly population. The studies investigated the ONSD change in children, the normal population, optic neuritis, and post-brain traumatic cases, with a limited number of patients [6,7,16,18,19,20]. In this study, we aimed to determine a standard ONSD value in healthy adults, who were 65 years of age or older and had not previously been diagnosed with a disease that could increase ICP.

## 2. Materials and Method

The study was approved by the University of Health Sciences Antalya Education and Research Hospital Clinical Research Ethics Committee (Antalya, Turkey), with the registry number 14/14, on 30 May 2019.

This prospectively-designed study was conducted in the emergency department of a tertiary hospital. The study included volunteers aged 65 years or older, who were admitted to the emergency department and had none of the exclusion criteria. The exclusion criteria were as follows: (1) being <65 years old; (2) having an ophthalmic disease (such as glaucoma, uveitis) that may affect CSF pressure or using associated medications; (3) having ophthalmic diseases, such as tumors, traumas, optic neuritis, or ocular prostheses; (4) being poisoned by drugs or substances; (5) having electrolyte disorders (such as hyponatremia or hypernatremia, hypocalcemia or hypercalcemia); (6) havingendocrine disorders (such as hypoglycemia or hyperglycemia, hypothyroidism or hyperthyroidism); (7) having had a previous intracranial event (history of ischemic or hemorrhagic stroke, subarachnoid hemorrhage, or central nervous system infection); (8) having a history of a brain tumor; (9) having a history of epilepsy; (10) having a history of Alzheimer’s disease; (11) having deficits on neurological examination; and (12) not giving consent to the study. Signed consent was obtained from all volunteers. All patients underwent physical, neurological and ophthalmologic examinations. The patients’ age, gender, Glasgow Coma Scale (GCS) score, past medical history characteristics, vital signs, physical examination findings, and right and left ONSD measurements were recorded.

Emergency physicians (EP) with at least three years of work experience in the emergency department and the ability to use USG in the management of patients participated in the study. All ocular ultrasonography examinations were performed by two EPs, with more than 3 years of experience in USG. Moreover, a total of 2 h of training, 1 h of theoretical and 1 h of practical training, relating to the measurement of ONSD by ocular USG was provided to the EPs, who participated in the study. All EPs practiced on 10 to 15 patients over a three-week period, before the study started.

### 2.1. Measurement of the Optic Nerve Sheath Diameter

All ONSD measurements were performed by two EPs in B-mode using the Mindray (Germany) ultrasound device, with a 7.5MHz linear array probe, orbital imaging settings and a high-resolution optimization setting. The procedure previously described in the literature was used for ONSD screenings [19,20,21]. The patients were examined with their eyelids closed, in a lying position. The head was elevated at 20°–30° to avoid any pressure on the eye, and the volunteers were asked to keep their eyes in the middle position and to suppress their eye movements. An ultrasound gel was applied to the surface of each eyelid. First, both eyes were scanned along the vertical and horizontal planes, then the optic disc of the eye was examined, and ONSD measurements were performed using hypoechoic lines, 3 mm proximal to the optic disc, as references (Figure 1). In order to minimize interobserver variation, the same USG device was used on each volunteer, each eye was examined 3 times, and the averages of these values were then calculated and recorded.

### 2.2. Statistical Analysis

All data were analyzed using the software SPSS Statistics Version 21.0 for Windows. Categorical data are presented as percentages. Numerical data are presented as the mean ± standard deviation, as well as the median (min–max), and the assumption of normality was tested using the Kolmogorov–Smirnov test. Categorical variables were analyzed using an χ^2^ test; parametric data were analyzed using the student t test and ANOVA test. Nonparametric data were analyzed using Mann–Whitney and Kruskal–Wallis tests. *p* < 0.05 was considered statistically significant.

The right and left ONSD values, mean ONSD values and ONSD differences were compared, according to the genders of the participants. The mean ONSD value was calculated as = (right ONSD + left ONSD)/2, and ONSD difference was calculated as = wider measured ONSD—narrower measured ONSD. The patients were divided into 3 groups, according to their ages. The age groups were assigned as follows: Group 1: 65–74 years of age, Group 2: 75–84 years of age and Group 3: 85 years of age or older [22]. The ONSDs and the ONSD difference between the right and left eyes of Group 1, Group 2 and Group 3 were compared.

## 3. Results

The study included 195 volunteers. There were 103 male and 92 female patients. The mean age was 75 ± 7 years (65–95 years). The volunteers’ right ONSD was 4.15 ± 0.70 mm (min: 2.2, max: 6.5), and their left ONSD was 4.18 ± 0.70 mm (min: 2.3, max: 5.9). The difference between the ONSD of the right and left eyes was 0.16 ± 0.18 mm (min: 0, max: 0.8). The mean ONSD of the right and left eyes was 4.16 ± 0.69 (min: 2.2, max: 6.2).

There was no difference between genders in terms of the right ONSD, left ONSD, mean ONSD and ONSD difference between the right and left eyes (*p* > 0.05) (Table 1).

There was no correlation between age and ONSD and ONSD difference (Table 2).

The age groups and ONSD were compared, and no difference was identified between the groups (Table 3).

## 4. Discussion

There is no consensus on the abnormal ONSD cut-off value that indicates increased ICP. It has been shown that ONSD increases with age in children, and that the rate of increase in the first few years of life is the highest [15,16]. It is not known whether or not ONSD varies beyond childhood and what its normal value is in people aged 65 years or older, known as the elderly population. In a study on healthy Pakistani adults, with a mean age of 31.08 ± 5.90 years, the median ONSD of the right eye was measured at 4.84 mm, and 95% of individuals had an ONSD in the range of 4.84–4.97 mm, while the median ONSD of the left eye was measured at 4.86 mm, and 95% of individuals had an ONSD in the range of 4.85–4.96 mm. In this study, no difference was found between ONSD, age and gender [23]. In a similar study, conducted on Chinese adults in Hong Kong, the mean ONSD was measured at 4.05 ± 0.19 mm. Again, there was no difference between genders in this study, and no significant correlation was found between age and ONSD [6]. In a study on healthy adults, the mean ONSD was measured at 4.8 ± 0.3 mm in males and 4.9 ± 0.2 mm in females, and there was no difference between genders. Again, in this study, there was no difference between every 10-year age range between the ages of 10 and 60 years in terms of ONSD, and the mean right ONSD was measured at 5.0 ± 0.1 mm, while the mean left ONSD was measured at 4.9 ± 0.2 mm in the 50–60-year-old age group [24]. In another study, where in the pre-and post-brain death ONSD values in neurologic patients, followed in the intensive care unit, were evaluated, the mean right and left ONSD values were measured at 4.5 mm in the control group (31 patients without central nervous system damage or clinical signs of intracranial hypertension) [10]. In our study, the ONSD in healthy volunteers aged 65 years or older was evaluated, and the mean ONSD was 4.16 ± 0.69 mm. At the same time, it was found that the ONSD was not different between groups, when the participants were divided into three groups, according to their age. When the correlation between ONSD and age was evaluated, based on this result, there was no correlation between them. Similar to other studies, it was found that the ONSD did not vary between genders, and that the mean ONSD of the right and left eyes was 4.15 ± 0.64 mm in females and 4.16 ± 0.74 mm in males.

While there is no difference between the mean ONSD, age and gender in the studies, it is seen that the ONSD range is very wide. In one study, the ONSD range was 3.7 mm to 4.7 mm in healthy adults [6], while in another study, it was measured in the range of 4.5–5.2 mm, in males, and in the range of 4.7–5.1 mm in females [24]. In another study, the right ONSD was measured in the range of 3.20–4.90 mm, while the left ONSD was measured in the range of 3.20–4.80 mm [25]. In our study, the right ONSD was in the range of 2.2–6.5 mm, while the left ONSD was in the range of 2.3–5.9 mm. These results suggest that ONSD measurement within these ranges may be insufficient for defining ICP increase. In such a case, the calculation of the difference between both ONSDs, as well as the measurement of ONSD, may be useful in determining ICP. Indeed, in a study investigating the efficacy of ONSD measurement in determining ICP increase, the mean ONSD difference was 0.97 ± 0.5 mm in the group of patients with an intracranial pathology, as shown in a CT scan, while it was 0.45 ± 0.4 mm in the group with a normal CT scan [4].In another study, the ONSD difference was 0.46 ± 0.55 mm in patients with acute ischemic stroke, while it was measured at 0.08 ± 0.06 mm in healthy adults [26]. In another study on children with head trauma, the measurement of ONSD was performed on CT scans. In this study, the mean ONSD difference between the left and right eyes was 0.47 ± 0.42 mm in patients with a pathology, as shown in a CT scan, while it was 0.27 ± 0.20 mm in patients with a normal CT scan. Moreover, the ONSD difference was 0.57 ± 0.51 mm in patients with a midline shift (MLS), while the ONSD difference was measured at 0.29 ± 0.24 mm in patients without MLS. Again, in this study, it was determined that the ONSD difference did not vary with age [17]. In addition, in another study, where in the ONSD values in neurologic patients, followed in the intensive care unit, were evaluated, both the right and the left ONSD differences were measured at 1.6 mm, before and after brain death [10]. In our study, the difference between the ONSDs of the left and right eyes was measured at 0.16 ± 0.18 mm. It was determined that there was no correlation between ONSD difference and age. The ONSD difference was similar between genders, and the ONSD difference between the left and right eyes was measured at 0.15 ± 0.17 mm in females and 0.18 ± 0.19 mm in males.

### Limitation

In our study, patients were divided into three subgroups, according to age. However, there was a difference between the age groups in terms of the number of patients. The lack of homogeneous distribution is the missing aspect of our study.

## 5. Conclusions

In conclusion, the ONSDs of both eyes do not vary with age in healthy adults aged 65 years or older. ONSD does not vary between genders. Furthermore, the ONSD measurement range is wide, and this should be taken into consideration when determining ICP. In addition, there is a difference between the ONSDs of the right and left eyes in healthy adults aged 65 years or older. The calculation of this difference, which does not vary with age or gender, can be used to determine ICP increase. However, more extensive studies are needed.

## Figures and Tables

**Figure 1 medicina-55-00413-f001:**
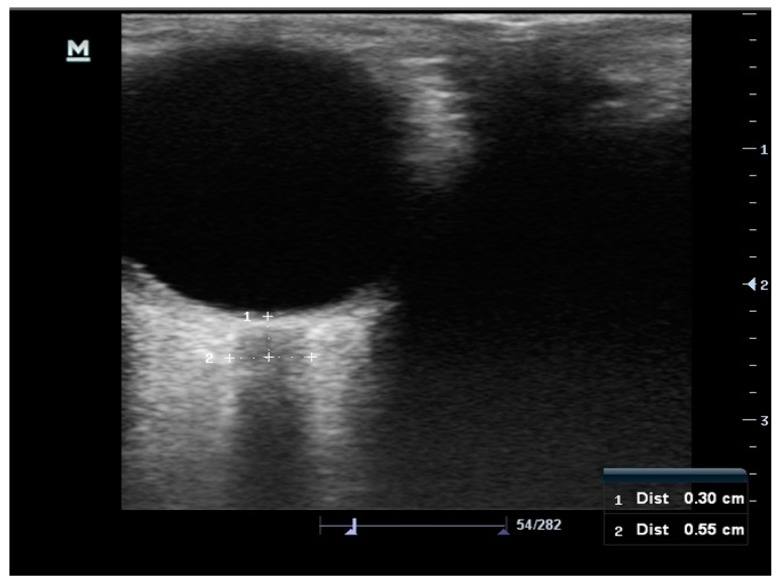
Theoptic nerve sheath diameter (ONSD) measurements were performed using hypoechoic lines, 3 mm proximal to the optic disc, as references.

**Table 1 medicina-55-00413-t001:** The optic nerve sheath diameter, according to sex.

Sex	Right ONSD	Left ONSD	Mean ONSD	ONSD Difference
	Mean ± SD (mm)	Mean ± SD (mm)	Mean ± SD (mm)	Mean ± SD (mm)
	Min–Max (mm)	Min–Max (mm)	Min–Max (mm)	Min–Max (mm)
P	0.831	0.894	0.883	0.236
Female (*n* = 92)	4.13 ± 0.65	4.17 ± 0.64	4.15 ± 0.64	0.15 ± 0.17
	2.2–5.9	2.3–5.8	2.3–5.8	0–0.7
Male (*n* = 103)	4.16 ± 0.74	4.18 ± 0.75	4.16 ± 0.74	0.18 ± 0.19
	2.3–6.5	2.7–5.9	2.7–6.2	0–0.8

Student t test, ONSD: optic nerve sheath diameter; P: significance level of the test.

**Table 2 medicina-55-00413-t002:** Correlation between age and the optic nerve sheath diameter.

**Age**		**Right ONSD**	**Left ONSD**	**ONSD Difference**
	R	−0.059	−0.070	0.022
	P	0.410	0.333	0.765

Pearson correlation test, ONSD: optic nerve sheath diameter; R: Pearson correlation coefficient; P: significance level of the test.

**Table 3 medicina-55-00413-t003:** Mean and standard deviation of the optic nerve sheath diameter in different age groups.

Age Groups	Right ONSD	Left ONSD	Mean ONSD	ONSD Difference
	Mean ± SD (mm)	Mean ± SD (mm)	Mean ± SD (mm)	Mean ± SD (mm)
	Min–Max (mm)	Min–Max (mm)	Min–Max (mm)	Min–Max (mm)
	%95 CI (LB–UB)	%95 CI (LB–UB)	%95 CI (LB–UB)	%95 CI (LB–UB)
P	0.981	0.907	0.964	0.522
65–74 (*n* = 104)	4.15 ± 0.69	4.20 ± 0.71	4.17 ± 0.69	0.15 ± 0.17
	2.4–5.9	2.6–5.9	2.5–5.8	0–0.7
	4.02–4.29	4.06–4.33	4.03–4.30	0.1–0.2
75–84 (*n* = 68)	4.14 ± 0.71	4.15 ± 0.71	4.14 ± 0.70	0.17 ± 0.18
	2.2–6.5	2.3–5.9	2.3–6.2	0–0.8
	3.97–4.31	3.98–4.32	3.97–4.31	0.1–0.2
>85 (*n* = 23)	4.12 ± 0.72	4.18 ± 0.67	4.15 ± 0.68	0.17 ± 0.19
	2.3–5.3	2.7–5.2	2.7–5.2	0–0.8
	3.81–4.43	3.89–4.47	3.85–4.44	0.1–0.2

ANOVA test, ONSD: optic nerve sheath diameter; P: significance level of the test; LB: lower bound; UP: upper bound.

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
