# Peer review of "The Measurement of Elderly Volunteers’ Optic Nerve Sheath Diameters by Ocular Ultrasonography"

_medicina, 2019, doi:10.3390/medicina55080413_

Round 1

Reviewer 1 Report

This is an interesting article dealing with the ONSD reference range in healthy elderly subjects.

Unfortunately there are several flaws:

Several studies in the current literature report the utility of A scan standardized echography to evaluate the ONSD,

these articles should be cited and discussed,

e.g.

1) Comment on "Optic Nerve Sheath Diameter Ultrasound Evaluation in Intensive Care Unit: Possible Role and Clinical Aspects in Neurological Critical Patients' Daily Monitoring".

De Bernardo M, Rosa N.

Biomed Res Int. 2018 Sep 16;2018:6154357

2) Optic nerve sheath diameter measurement in patients with idiopathic normal-pressure hydrocephalus.

De Bernardo M, Rosa N.

Eur J Neurol. 2018 Feb;25(2):e24.

Tables 1-3: please specify what P means, and how was the statistical analysis performed, for each table

Discussion: the age subgroups are not homogeneous. The authors should discuss this point in the text among one of the limitations of this study.

Author Response

Response to Reviewer 1 Comments

Dear Reviewer 1,

We are thankful to you for your recommends.

We revised our manuscript as your and the reviewer 2's recommends.

We listed all the changes we made below.

We hope you will accept our manuscript.

Sincerely yours,

Corresponding author.

Point 1: Open Review

English language and style

( ) Extensive editing of English language and style required 
(x) Moderate English changes required 
( ) English language and style are fine/minor spell check required 
( ) I don't feel qualified to judge about the English language and style 

Response1: The MDPI English Editing Service edited all of our manuscript. The changes can be seen in the word document and throughout the document.

Point 2:

Yes

Can be   improved

Must be   improved

Not   applicable

Does the   introduction provide sufficient background and include all relevant   references?

( )

(x)

( )

( )

Is the research   design appropriate?

( )

(x)

( )

( )

Are the   methods adequately described?

( )

(x)

( )

( )

Are the   results clearly presented?

( )

(x)

( )

( )

Are the   conclusions supported by the results?

( )

(x)

( )

( )

Response 2: We tried to improve all sections of our manuscript. The changes can be seen in the word document and throughout the document.

Comments and Suggestions for Authors

This is an interesting article dealing with the ONSD reference range in healthy elderly subjects.

Unfortunately there are several flaws:

Point 3: Several studies in the current literature report the utility of A scan standardized echography to evaluate the ONSD,

these articles should be cited and discussed,

e.g.

1) Comment on "Optic Nerve Sheath Diameter Ultrasound Evaluation in Intensive Care Unit: Possible Role and Clinical Aspects in Neurological Critical Patients' Daily Monitoring".

De Bernardo M, Rosa N.

Biomed Res Int. 2018 Sep 16;2018:6154357

2) Optic nerve sheath diameter measurement in patients with idiopathic normal-pressure hydrocephalus.

De Bernardo M, Rosa N.

Eur J Neurol. 2018 Feb;25(2):e24.

Response 3: We cited both two articles in introduction section (Reference numbers are 9 and 12). Furthermore, we cited an additionally reference numbered 10 (Toscano M, Spadetta G, Pulitano P, Rocco M, Di Piero V, Mecarelli O, et al. Optic Nerve Sheath Diameter Ultrasound Evaluation in Intensive Care Unit: Possible Role and Clinical Aspects in Neurological Critical Patients' Daily Monitoring. BiomedResInt. 2017;2017:1621428. doi: 10.1155/2017/1621428. Epub 2017 Mar 21.) Page 2 Line 60-69, Page 6 Line 177-180, 205-208.

Point 4: Tables 1-3: please specify what P means, and how was the statistical analysis performed, for each table

Response 4: We specified what P means and how the statistical analysis was performed, for each table.  Page 5 under the tables 1-3.

Point 5: Discussion: the age subgroups are not homogeneous. The authors should discuss this point in the text among one of the limitations of this study.

Response 5: We added a limitation section and mentioned the point that the age subgroups are not homogeneous. Page 7 Line 213-216.

Reviewer 2 Report

This paper deals with a really interesting topic, but this technique has several drawbacks, as it has been clearly stated by several papers in the international literature (among the others: 1) Ocular Ultrasound Assessment to Estimate the Risk of Increased Intracranial Pressure after Traumatic Brain Injury in Prehospital Setting. De Bernardo M, Vitiello L, Rosa N. Prehosp Emerg Care. 2019 Jan 11:1-2. doi: 10.1080/10903127.2019.1568652.

2) Comment on 'Invasive and noninvasive means of measuring intracranial pressure: a review'.  De Bernardo M, Rosa N. Physiol Meas. 2018 Jun 1;39(5):058001.

3) Optic nerve ultrasonography for evaluating increased intracranial pressure in severe preeclampsia.

De Bernardo M, Vitiello L, Rosa N. Int J Obstet Anesth. 2019 May;38:147.)

These topics should be addressed and discussed

The kind of test used for the different variables should be specified in the  tables.

Paper needs to be edited by a native English speaker.

Author Response

Response to Reviewer 2 Comments

Dear Reviewer 2,

We are thankful to you for your recommends.

We revised our manuscript as your and the reviewer 1's recommends.

We listed all the changes we made below.

We hope you will accept our manuscript.

Sincerely yours,

Corresponding author.

Point 1: Open Review

English language and style

( ) Extensive editing of English language and style required 
(x) Moderate English changes required 
( ) English language and style are fine/minor spell check required 
( ) I don't feel qualified to judge about the English language and style 

Response1: The MDPI English Editing Service edited all of our manuscript. The changes can be seen in the word document and throughout the document.

Point 2:

Yes

Can be   improved

Must be   improved

Not   applicable

Does the   introduction provide sufficient background and include all relevant   references?

( )

(x)

( )

( )

Is the   research design appropriate?

(x)

( )

( )

( )

Are the   methods adequately described?

( )

(x)

( )

( )

Are the   results clearly presented?

( )

(x)

( )

( )

Are the   conclusions supported by the results?

( )

(x)

( )

( )

Response 2: We tried to improve all sections of our manuscript. The changes can be seen in the word document and throughout the document.

Comments and Suggestions for Authors

Point 3: This paper deals with a really interesting topic, but this technique has several drawbacks, as it has been clearly stated by several papers in the international literature (among the others: 1) Ocular Ultrasound Assessment to Estimate the Risk of Increased Intracranial Pressure after Traumatic Brain Injury in Prehospital Setting. De Bernardo M, Vitiello L, Rosa N. Prehosp Emerg Care. 2019 Jan 11:1-2. doi: 10.1080/10903127.2019.1568652.

2) Comment on 'Invasive and noninvasive means of measuring intracranial pressure: a review'.  De Bernardo M, Rosa N. Physiol Meas. 2018 Jun 1;39(5):058001.

3) Optic nerve ultrasonography for evaluating increased intracranial pressure in severe preeclampsia.

De Bernardo M, Vitiello L, Rosa N. Int J Obstet Anesth. 2019 May;38:147.)

These topics should be addressed and discussed

Response 3: We cited these three articles in introduction section (Reference numbers are 11, 13 and 14). Page 2 Line 60-69. 

Point 4: The kind of test used for the different variables should be specified in the  tables.

Response 4: The kind of test used for the different variables was specified in the tables. Page 5 under the tables 1-3.

Point 5: Paper needs to be edited by a native English speaker.

Response 5: The MDPI English Editing Service edited all of our manuscript. The changes can be seen in the word document and throughout the document.

Round 2

Reviewer 1 Report

The authors modified the text according to the previous suggestions